# Sound Damage Detection of Bridge Expansion Joints Using a Support Vector Data Description

**DOI:** 10.3390/s23073564

**Published:** 2023-03-29

**Authors:** Junshi Li, Caiqian Yang, Jun Chen

**Affiliations:** 1School of Civil Engineering, Xiangtan University, Xiangtan 411105, China; junshili710@smail.xtu.edu.cn (J.L.); chenjun0325@126.com (J.C.); 2School of Civil Engineering, Southeast University, Nanjing 211189, China

**Keywords:** modal bridge expansion joint, damage detection, sound signal, wavelet packet energy ratio, support vector data description

## Abstract

A novel method is proposed for the damage identification of modal bridge expansion joints (MBEJs) based on sound signals. Two modal bridge expansion joint specimens were fabricated to simulate healthy and damaged states. A microphone was used to collect the impact signals from different specimens. The wavelet packet energy ratio of the sound signal was used to identify the difference in specimen state. Firstly, the wavelet packet energy ratio was used to establish the feature vectors, which were reduced dimensionality using principal component analysis. Subsequently, a support vector data description model was established to detect the difference in the signals. The identification effects of three parameter optimization methods (particle swarm optimization, genetic algorithm optimization, and Bayesian optimization) were compared. The results showed that the wavelet packet energy ratio of sound signals could effectively distinguish the state of the support bar. The support vector data description of Bayesian optimization worked best, and the proposed method could successfully detect damage to the support bar of MBEJs with an accuracy of 99%.

## 1. Introduction

Bridge expansion joints are an essential part of bridges, which enable vehicles to pass over the bridge smoothly and meet the needs of bridge deformation. However, the life of bridge expansion joints is often shorter than the bridge itself. The main reasons are the repeated expansion of bridge expansion joints and vehicle impact. Many scholars have studied bridge expansion joints, including the position of damage, dynamic characteristics, and other factors [1,2,3,4,5,6,7,8,9,10,11,12]. Modal bridge expansion joints (MBEJs) are common bridge expansion joints with a simple design, good water resistance, and confinement. They are suitable for most bridges [13,14]. The support bar of an MBEJ is the component that mainly undertakes the expansion task, which is highly susceptible to damage. Damage identification is difficult because the support bar is inside the MBEJ. In real bridges, the damaged state of the support bar is often detected manually. However, manual checking is influenced by subjective factors. Therefore, there is an urgent need for a suitable method to identify damage to the support bar.

Sound is an acoustic wave produced by the vibration of an object. Sound transmission has the advantage of being omnidirectional and requiring no contact. It is suitable to collect information without visibility. An impact sound is produced when a vehicle drives over an MBEJ. The sound indicates the state of the interior of the MBEJ, especially the support bar of the MBEJ. 

Many scholars have used sound signals for damage and fault identification. Zhuo et al. [15] proposed an online diagnostic procedure for steel truss structures. The program’s primary method was extracting features from the sound signal associated with a damaged bolt connection. The sound signals were collected in a microphone array. The time domain and the wavelet packet energy of the sound signal defined the features. A support vector machine was used to identify bolt loosening. Steering response power values of coordinates in an offline database obtained damage localization. Krause et al. [16] proposed an acoustic-emission-signal-based method for rotor blade damage detection for wind turbines. The method extracted the features of rotor blade damage signals from spectrograms. Decision trees were chosen as the classification method. The method could detect the damage state associated with full-scale fatigue testing without false detection. Suman et al. [17] proposed an algorithm using the Kalman filter to reduce the noise and the Mel frequency cepstral coefficient for fault identification. The method used the machinery’s vibration and acoustic signals to identify mechanical faults in vehicles. Wan et al. [18] designed an acoustic-based pipeline monitoring method. The method compared two sound characteristics (the Mel frequency cepstral coefficient and the linear predictive cepstral coefficient). The features were input into the Euclidean inverse spectral distance and one class of support vector machines. The results showed that the Mel frequency cepstral coefficient is more sensitive to damage and both classifiers had good results. Arora et al. [19] proposed a method for damage detection using acoustics. The method used the acoustic pressure response measured by a microphone for vibroacoustic modal analysis to obtain acoustic characteristics. The acoustic characteristics were used to calculate the variation of the vibroacoustic flexibility matrix for damaged and healthy structures. The differences between the vibroacoustic flexibility matrices were used to detect the damaged location and state. A plate structure was used to verify the feasibility of the method. Jha et al. [20] proposed a multi-class support-vector-machine-based fault classification method for the fault diagnosis of bearings. The method transformed the one-dimensional vibration signal into a two-dimensional grayscale image and extracted feature vectors from the image. The feature vectors were used to train a multi-class support vector machine. The results showed that the method could classify the location and degree of faults with high robustness. Aymerich et al. [21] utilized nonlinear acoustic techniques to detect impact damage in composite plates. Chen et al. [22] proposed a damage detection method for wind turbine blades based on acoustic signals. The signal was filtered, and wavelet packet transform was performed on the processed signal to obtain the wavelet energy. The wavelet energy ratio was used to train the support vector data description (SVDD) model. An improved incremental learning method was used to adaptively update the SVDD. Finally, the feasibility of the method was verified using the measured data. 

At the same time, sound signals are also used in other applications. Guo et al. [23] proposed a shock acoustic signal processing technique based on Gaussian modeling and an improved extreme learning machine method. Healthy, pest-infested, and germinated wheat was classified by the method with an accuracy of 90%. Wang et al. [24] proposed a new impact-based method using analytical modeling and numerical simulation. The method used the virtual material method and layering theory to model the bolted connections equivalently. The acoustic radiation pattern method was obtained for the impact of sound pressure levels. A numerical model was proposed for acoustic–structural coupling. The method could quickly evaluate bolt preload in industrial environments through cheap acoustic tests. Liu et al. [25] designed a sound monitoring system in order to prevent damage to underground pipelines caused by construction. The system extracted the acoustic features of sound signals. With good results, the random forest classifier classified the construction sounds of electric hammers, road cutters, excavator breakers, and environmental noise. Xie et al. [26] constructed a new feature set. The feature set extracts an aggregated feature set of several acoustic features from the acoustic signal, such as short-time energy, short-time cross zero rates, and the Mel frequency cepstral coefficient, and visual features such as gradient histograms. The feature sets were trained with K-nearest neighbors, a support vector machine, and four other classifiers for comparison. The research provided the theoretical basis for engineering applications. Papandrea et al. [27] proposed a method for diagnosing surface roughness. The method extracted the spectral energy of each frame of the acoustic signal and then performed dimensionality reduction using principal component analysis and input to be trained for surface roughness diagnosis. The accuracy of the classification was 100%. Luo et al. [28] proposed a method for testing the natural frequency of steel based on acoustic signals. The periodogram method analyzed the acquired sound signals to obtain the power spectral density curve. The power spectral density curves obtained the natural frequencies of steel. Finally, the measured data and simulations verified the feasibility of the method. The method could identify the natural frequencies of steel before and after damage. In addition, acoustic emission signals generated from inside objects are also widely used [29,30,31,32,33,34].

In past studies, sound signals have been used to detect and assess the state of objects in an intact way. Sound transmission has the advantage of being omnidirectional and requiring no contact, allowing for the characterization of the state inside an object. The identification of damage to the support bar of an MBEJ is a difficult task that requires the determination of the state of the interior of the MBEJ. Therefore, for scientific and accurate identification of the support bar, this paper proposed a new method for damage detection of the support bar of MBEJs using sound signals. Firstly, the sound signals were collected through a microphone. Subsequently, wavelet packet transform was applied to decompose the sound signal to obtain the wavelet energy ratio, and the feature vector was obtained from the wavelet energy ratio. The dimensionality of the feature vector was reduced using principal component analysis. Then, the damage to the support of the MBEJ was identified using the SVDD. Finally, three methods of parameter optimization for the SVDD were compared. The results showed that the proposed method could effectively detect damage to the support of the MBEJ.

The rest of the paper is organized as follows: Section 2 describes the sound-signal-based bridge expansion joint damage identification method; Section 3 presents the details of the experiments; and Section 4 discusses the results of the experiments. Finally, the paper is summarized in Section 5.

## 2. Theoretical Background

In engineering, most structures are healthy, and data on damaged structures are often difficult to obtain. There are two prominent cases of acquired data: the first is that the acquired database contains mostly healthy structural data and a small percentage of damaged data. The second is that the acquired database contains data on all healthy structures. The traditional support vector machine needs to be improved for damage identification of MBEJs because it requires sufficient data on health and damage. SVDD is a single classification algorithm that can build a database with health data. This paper uses SVDD to perform the study analysis. Figure 1 shows the flowchart of the proposed method. The initial step is to collect and process the sound signal. Then, the wave packet energy ratio of the processed signal is calculated and used as a feature vector. The wave packet energy can reflect the signal’s energy percentage in different frequency bands. Next, the dimensionality of feature vectors is reduced using principal component analysis, which is used to compute the SVDD model. In the training of SVDD, the selection of key parameters is essential. This paper uses three parameter optimization methods to select the best parameters for SVDD. Finally, three methods of SVDD parameter optimization are selected for comparison. Section 2.1 discusses the method of feature extraction (wavelet packet transform); Section 2.2 discusses three methods of parameter optimization; and Section 2.3 reviews the methods of SVDD.

### 2.1. Feature Extraction

Wavelet transform can obtain the signal’s time domain and frequency characteristics, unlike the conventional Fourier transform [35]. However, wavelet transform needs better frequency resolution in the high-frequency band and better time resolution in the low-frequency band. Wavelet packet transform is a new analysis method based on wavelet transform. Wavelet packet transform can accurately distinguish the high-frequency part of the vibration signal, unlike wavelet transform. Wavelet packet transform can divide the signal into the frequency band range. Wavelet packet transform is calculated as follows:(1)$$d_{i,j,2m}=\sum_{k}h(k-2i)d_{k,j+1,m}$$
(2)$$d_{i,j,2m+1}=\sum_{k}g(k-2i)d_{k,j+1,m}$$

*d_i,j,m_* is the node *i* wavelet coefficient of the node *m* in layer *j. h(k)* and *g(k)* are the multi-resolution analysis’ orthogonal mirror filter’s low-pass and high-pass filter coefficients. The energy *E_i,j_* of wavelet packet decomposition at different frequency bands is calculated as follows:(3)$$E_{i,j}=\sum_{k=1}^{N}{\left|d_{i,j}(k)\right|}^{2},j=0,1,\cdots 2^{i}-1$$

The feature vector consists of wavelet packet energy ratio *P_i,j_*, which is calculated as follows:(4)$$P_{i,j}=\frac{E_{i,j}}{\sum_{j=0}^{2^{i}-1}E_{i,j}}$$

Principal component analysis (PCA) is used to optimize the feature vectors, removing possible redundant information and reducing the algorithm’s computational overhead [36]. PCA is a multivariate statistical analysis method. The core theory of PCA is to reduce high-dimensional features to a few essential features.
(5)$$Z=[z_{1},z_{2},\ldots ,z_{n}]$$

Equation (5) is the feature vector after principal component analysis. The *Z* of PCA is input into the SVDD model. In order to retain more fault features, the contribution rate of the appropriate principal components needs to be selected.

### 2.2. Parameter Optimization

In SVDD, the model’s effect is determined using the penalty parameter C and the kernel parameter. C controls the trade-off between the hypersphere volume and the model classification error. The kernel width parameter controls the shape of the hypersphere. Therefore, it is crucial to select the most appropriate parameters accurately. Common hyperparametric optimization algorithms include Bayesian optimization (BO), particle swarm optimization (PSO), genetic algorithms (GAs), and others.

#### 2.2.1. Bayesian Optimization

Bayesian optimization (BO) obtains preliminary information from the existing parameter choices, and it continuously updates the objective function by a given hyperparameter to guide the next parameter choice [37,38]. The optimization process is as follows:(6)$$p\left(f\;|R_{1:t}\right)=\frac{p\left(R_{1:t}|f\right)p(f)}{p\left(R_{1:t}\right)}$$
(7)$$R_{1:t}=\left\{\left(x_{1},y_{1}\right),\left(x_{2},y_{2}\right)\cdots \left(x_{t},y_{t}\right)\right\}$$
(8)$$y_{t}=f\left(x_{t}\right)+\varepsilon_{t}$$
where *f* represents the objective function; *y_t_* represents the observed value at step *t*; *x_t_* represents the hyperparameter of step *t*; *ε**t* represents the observation error; *R*_1:*t*_ represents the summary of observations from the previous *t* steps; the likelihood distribution is represented by *p*(*R*_1:*t*_|*f*); *p*(*f*) represents the prior distribution of *f*; *p*(*f*) is considered as the state assumption of the objective function; and *p*(*f*|*R*_1:*t*_) represents the posterior distribution of the objective function.

The Gaussian process is used as a probabilistic surrogate model, determined by the mean and covariance functions.
(9)$$f(x)\sim \mathrm{GP}\left(m(x),k\left(x,x^{\prime }\right)\right)$$

*M* (*x*) is the mean function and *k* (*x*, *x*′) is the covariance function.
(10)m(x)=E[f(x)]
(11)$$k\left(x,x^{\prime }\right)=E\left[(f(x)-m(x))\left(f\left(x^{\prime }\right)-m\left(x^{\prime }\right)\right)\right]$$

#### 2.2.2. Particle Swarm Optimization

Particle swarm optimization (PSO) is a population-based heuristic algorithm [39]. The investigation of group social behavior and intelligence inspires technology. This equation updates the velocity of each particle:(12)$$v_{i}(t+1)=wv_{i}(t)+c_{1}r_{1}\left[{\hat{x}}_{i}(t)-x_{i}(t)\right]+c_{2}r_{2}\left[g(t)-x_{i}(t)\right]$$
where *x_i_*(*t*) describes the position of the particle. ${\hat{x}}_{i}(t)$ denotes the individual best solution of the particle. *r*_1_ and *r*_2_ illustrate the random numbers that are uniformly distributed in the interval. *g*(*t*) represents the optimal solution for the population. *c*_1_ represents the cognitive coefficient. *c*_2_ illustrates the social coefficient. *c*_1_ and *c*_2_ generally take the values [0–4]. *w* is the inertia weighting coefficient. A larger *w* is good for global search and does not fall into the local optimum. A smaller *w* is good for local search and can converge quickly to obtain the optimal solution. The general value is [0.4–2].

The following equation calculates the position of the following particle:(13)$$x_{i}(t+1)=x_{i}(t)+v_{i}(t+1)$$

#### 2.2.3. Genetic Algorithm

A genetic algorithm (GA) is an adaptive heuristic search algorithm that aims to simulate the process of gene selection and natural selection in the theory of biological evolution [40]. A GA uses natural selection, hybridization, and other means to achieve population evolution. It can search for optimal solutions randomly and quickly.

### 2.3. Support Vector Data Description 

In the damage monitoring of bridge expansion joints, the amount of healthy and damaged data needs to be more balanced, and damaged data are challenging to obtain. Therefore, monitoring expansion joints is a single-classification problem. Support vector data description (SVDD) is a single classification method that aims to find the smallest possible hypersurface to enclose more of the target data [41,42,43]. It is suitable for the damage monitoring of support bars.
(14)$$\min R^{2}+C\sum_{i=1}^{n}\xi_{i}$$
(15)$$\;s.t.\;{\left\|x_{i}-a\right\|}^{2}\leq R^{2}+\xi_{i},\xi_{i}>0\;\forall i$$
where *ɑ* is the center of the hypersphere and *R* is the radius of the hypersphere. The variable *ζ**i* is the slack variable. C is the penalty parameter. *x*_i._ is the test point. The Lagrange multipliers *α**i* ≥ 0 and *γ**i* ≥ 0 are used to denote (1):(16)$$L=R^{2}+C\sum_{i}\xi_{i}-\sum_{i}\gamma_{i}\xi_{i}-\sum_{i}\alpha_{i}\left\{R^{2}+\xi_{i}-\left({\left\|x_{i}\right\|}^{2}-2a\cdot x_{i}+\|a{\|}^{2}\right)\right\}$$

Taking partial derivatives of *ɑ*, *R*, *ζ*_*i*_:(17)$$\sum_{i}\alpha_{i}=1$$
(18)$$a=\sum_{i}\alpha_{i}x_{i}$$
(19)$$0\leq \alpha_{i}\leq C$$

The objective function can be written as follows:(20)$$L=\sum_{i}\alpha_{i}\left(x_{{}_{i}}\cdot x_{{}_{{}_{i}}}\right)-\sum_{i,j}\alpha_{i}\alpha_{j}\left(x_{i}\cdot x_{j}\right)$$

In this paper, the Gaussian kernel function is chosen to replace the inner product operation. The core parameters are as follows:(21)$$K\left(x_{i}\cdot x_{j}\right)=\exp\left(\frac{-{\left\|x_{i}-x_{j}\right\|}^{2}}{2\delta^{2}}\right)$$
where *δ* is the width parameter of the function; *K*(*x*_*i*_⋅*x*_*j*_) is the kernel function.

In addition, *R* is the distance between the center *ɑ* and any support vector *x_p_*, 0 ≤ *α**i* ≤ *C*.
(22)$$R^{2}={\left\|x_{p}-a\right\|}^{2}=K\left(x_{p}\cdot x_{p}\right)-2\sum_{i}\alpha_{i}K\left(x_{i}\cdot x_{p}\right)+\sum_{i,j}\alpha_{i}\alpha_{j}K\left(x_{i}\cdot x_{j}\right)$$

For a test sample *z*, its distance from the center *ɑ* is calculated as follows:(23)$$d_{z}^{2}=\|z-a{\|}^{2}=K(z\cdot z)-2\sum_{i}\alpha_{i}K\left(z\cdot x_{i}\right)+\sum_{i,j}\alpha_{i}\alpha_{j}K\left(x_{i}\cdot x_{j}\right)$$

When *d* ≤ *R*, sample z is considered health data.

## 3. Experimental Details

In bridges, the concrete in the anchorage zone on both sides of the MBEJ connects the deck slab and the bottom of the MBEJ, and the deck slabs mostly overhang. However, it is the side girders on both sides of the MBEJ that are most restricted by the concrete in the anchorage zone and will not restrict the support bar, and the concrete at the bottom of the specimen will not restrict the support bar. Thus, the changes in boundary conditions at the bottom and on both sides of the MBEJ do not affect the target (support bar) identified for this test. Therefore, only enough concrete needs to be poured to confine the edge beams of the MBEJ, and no excessively long deck slabs need to be cast.

### 3.1. Experimental Materials

In order to verify the applicability of the proposed method, the following experiments were carried out in the Xiangtan University civil engineering laboratory. MBEJ, reinforcement, and concrete were used to fabricate specimens to simulate the project. Hengshui Boyun Rubber Products Company’s GQF-MZL160 MBEJs were used to make the specimens. The MBEJ was mainly composed of one center beam, two support boxes, two edge beams, two support bars, and two sealing rubber bands. The materials were Q345B steel, except for the two sealing rubber bands. The material properties are shown in Table 1. Two specimens were cast using concrete with a cubic compressive strength of 50 MPa, and they were cured for 28 days. The design of the concrete is shown in Table 2. The diameter of the stirrup was 10 mm and it was a hot-rolled ribbed steel bar (HRB400), and the other reinforcement was a 16 mm hot-rolled ribbed steel bar (HRB400). The details of the reinforcement of the specimen are shown in Figure 2. Specimen 2 is the MBEJ in a healthy state, and specimen 1 is the MBEJ after damage to the support bar. The length of the specimen was 1500 mm, the width was 1200 mm, and the height was 500 mm. The specimen and experimental details are shown in Figure 3.

### 3.2. Experimental Process

Due to the speed and safety of oxygen-cutting technology, the support bar of specimen 1 was cut using oxygen-cutting technology to simulate the actual fracture. In order to acquire the sound signal under the impact action, the specimen was excited by a hammer. The hammer excitation method was used to simulate the impact of vehicles on the MBEJ. Since the damaged part is located in the support bar, the sound generated by exciting the center beam in the upper part of the support bar is more representative of the characteristics of the support bar. Therefore, the excitation position was the center beam in the upper part of the support bar. A PM461 microphone, produced by Shenzhen Moro Zhiyuan Technology Company, was used to collect the sound signal of the hammer impact with a sampling frequency of 20 kHz. In order to ensure the signal integrity and the quietness of the test site, the experimental time was chosen to be at night. Therefore, the experiments do not consider noise. During the test, two 50 mm thick foam pads were placed at the bottom of the specimen to reduce the disturbance of the ground. Finally, all of the signals collected for health came from specimen 2. The damaged signals came from specimen 1, in which the support bar was broken. Figure 4 shows the two specimens of the experiment. Figure 5 shows the healthy and damaged support bars.

## 4. Results and Discussion

Figure 6a,b shows the excitation signals when the beam is damaged and healthy, respectively. Since the test was conducted at night and sufficient quietness was ensured, the time domain plots are smoother. The experiment obtained 600 datasets. The datasets have 500 healthy states and 100 damaged states. Daubechies wavelet (dbN) is a type of wavelet basis function. The characteristic of the dbN wavelet is that the order of vanishing moments becomes more significant as N increases. The larger the vanishing moment, the better the smoothness, the stronger the localization ability of the frequency domain, and the better the division of the frequency band. However, the amount of computation can increase significantly. Therefore, the db6 basis function was used to apply wavelet packet decomposition to the processed signal. 

A representative set of data was extracted from several hundred sets of data, and each set had the same characteristics. Figure 7 shows the results of this dataset’s four-layer and five-layer wavelet packet decomposition, including healthy and damaged states. Figure 7a,b shows that the frequency band division of the four-layer wavelet packet decomposition is not apparent. Before and after the damage changes, the wavelet packet energy ratio is less in numerical value, with the first frequency band increasing by only 15.96. The energy ratio of the five-layer wavelet packet decomposition changes significantly in numerical value before and after damage. The energy ratio increase in the first frequency band is 45.34. The energy ratio of the second, third, and fourth frequency bands is reduced to about two following damage. The five-layer wavelet packet decomposition is better than the four-layer, which responds to a significant change in damage. Therefore, this paper selected five-layer wavelet packet decomposition to extract the feature vectors. Figure 7c,d shows the results of the five-layer wavelet packet energy ratio.

Figure 8 illustrates the results of the cumulative contribution rate (CCR) using the PCA. The CCR increased from 97.84% to 99.52% when the number of principal components was increased from one to two. Subsequently, the CCR curve gradually flattens, and the amount of information contained in the principal components decreases. When the number of principal components was increased from 2 to 32, the CCR increased by only 0.48%. Therefore, the first two dimensions of the principal component analysis were used for SVDD to improve the computational efficiency.

The optimized sample data were normalized after optimization. Since damage data are more difficult to obtain in engineering, this study built a training set with healthy data. Four hundred healthy samples were used as training sets. The test sample contained 100 healthy and 100 damaged samples, in which the healthy samples were positive samples and the damaged ones were negative samples. The conventional SVDD method calculates the center and radius of the sphere in the positive state. The penalty parameter C is initially set to 20. The Gaussian kernel function is selected as the kernel function, and the kernel parameters are initially set to 15.

Figure 9 shows the training results of SVDD. Training created ten support vectors, which created a decision boundary. The accuracy rate of the test was 95%. Figure 10 shows the test results of the test set, where the green point is the positive sample and the orange point is the negative sample. The horizontal line in Figure 11 represents the radius of the hypersphere obtained by training the positive sample. The radius R of the hypersphere obtained was 0.8704. The test results conclude that SVDD divides the negative samples clearly, but identifying the positive samples requires more accuracy. Most of the positive samples were within the hypersphere radius and a few of the positive samples were outside the hypersphere radius R. There are two main reasons for this. The first reason is that the density distribution of the support vectors used for training is not uniform, resulting in less than reasonable training of the decision boundary. The second reason is that the penalty and kernel parameters are not suitable for the SVDD model. Therefore, parameter optimization with a suitable algorithm is required.

In SVDD, the choice of parameters determines the accuracy of the model. PSO has the advantages of fast convergence, few parameters, and a simple algorithm. The GA is flexible enough to solve a variety of complex optimization problems with high computational efficiency and can solve optimization problems in any dimension. BO is computationally efficient by building a probabilistic model from the past evaluation results of the objective function. All three methods can optimize the parameters of SVDD. Therefore, three parameter optimization methods (PSO, GA, and BO) were used to optimize the parameters of SVDD. As with the unoptimized SVDD, the feature vectors were still optimized using PCA with a training set of 400 undamaged samples. The test sample contained 100 healthy and 100 damaged samples, where the healthy samples were positive and the damaged samples were negative. Figure 12, Figure 13 and Figure 14 show the training and testing results of BO-SVDD. Training produced three support vectors, further expanding the boundaries and reasonably enclosing the samples. The accuracy of all three methods was 99%. The parameter-optimized SVDD improved the accuracy by 4% over SVDD. The optimized decision boundary encloses the non-destructive data that SVDD does not enclose. The optimized SVDD identifies all the negative samples in the test set. There are only two pieces of abnormal data in the positive samples tested due to the electromagnetic interference of the condenser microphone during the sampling process.

Table 3 shows the comparison results of SVDD, PSO-SVDD, GA-SVDD, and BO-SVDD under the same computer configuration. The cost is the penalty parameter weighing the hypersphere volume and misspecification rate. Gamma is the kernel parameter. R is the hypersphere radius. SVDD has the shortest time, but it does not have high accuracy since SVDD is not parameter-optimized. The SVDD of the optimized parameters all achieved high and similar accuracy. PSO-SVDD, GA-SVDD, and BO-SVDD all take longer than SVDD since the parameter optimization needs some time to find the optimal parameters. However, compared to the other three, BO has the shortest time. The reason is that BO requires only constant sampling. It will refer to the previous evaluation results to infer the optimal value of the function. There are also a few points to sample, so the time is short. For this experiment, BO is optimal with the same accuracy rate. 

## 5. Conclusions

This paper has proposed a method for damage identification for the support bar of MBEJs based on sound signals. The features were extracted from the sound signal using wavelet packet transform, and PCA reduced the dimensionality of the features. The SVDD model was built from the reduced dimensional data, and two specimens were valid for the method’s feasibility. Subsequently, three parameter-optimized SVDD models (PSO-SVDD, GA-SVDD, and BO-SVDD) were compared. The results showed that BO-SVDD was the optimal model and performed best regarding the training time. It achieved 99% accuracy and a 4% improvement on the accuracy of SVDD. Despite the short training time of SVDD, its accuracy needed to be higher. Therefore, the proposed BO-SVDD was more suitable for the requirement of online damage identification of MBEJs. The limitation of this study is that the method can only consider the state where the specimen does not expand. In later work, the method needs to consider the effect of noise when used on bridges. In the future, online and adaptive models will be further investigated and applied in engineering. This paper reports the preliminary results of the methodology, and the work represents a pilot study to assess the feasibility of the developed monitoring technology.

## Figures and Tables

**Figure 1 sensors-23-03564-f001:**
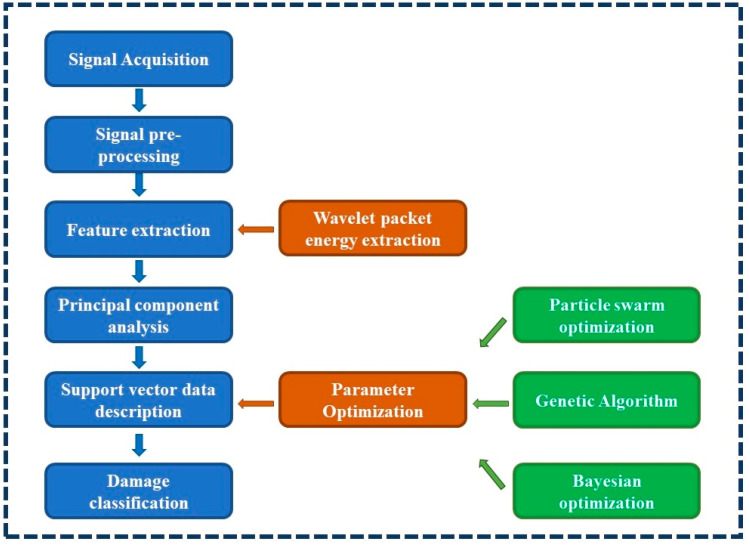
Flowchart of the method.

**Figure 2 sensors-23-03564-f002:**
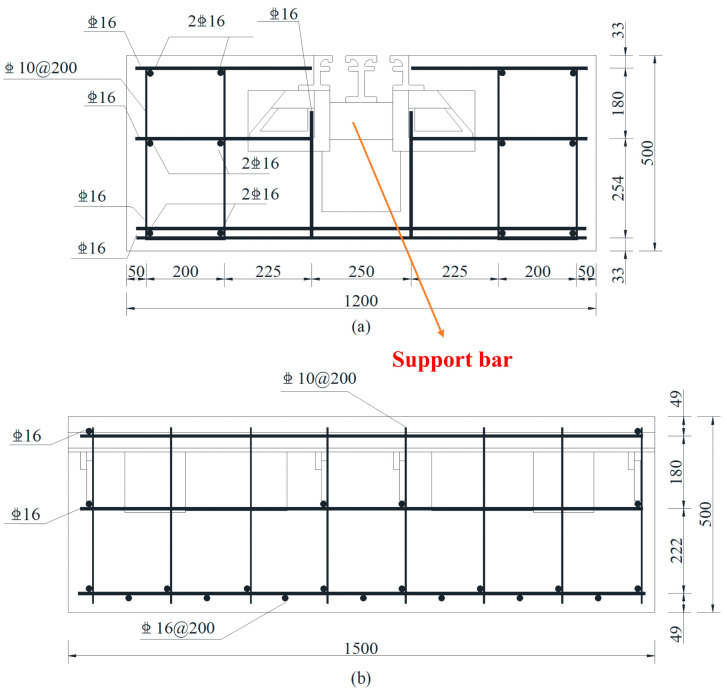
Details of the reinforcement of the specimen: (**a**) front view of the specimen; (**b**) side view of the specimen.

**Figure 3 sensors-23-03564-f003:**
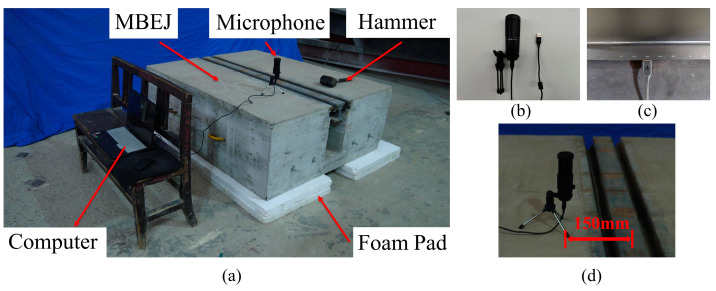
Experimental details: (**a**) experimental process; (**b**) microphone; (**c**) interface for computer and microphone connection; (**d**) location of the microphone.

**Figure 4 sensors-23-03564-f004:**
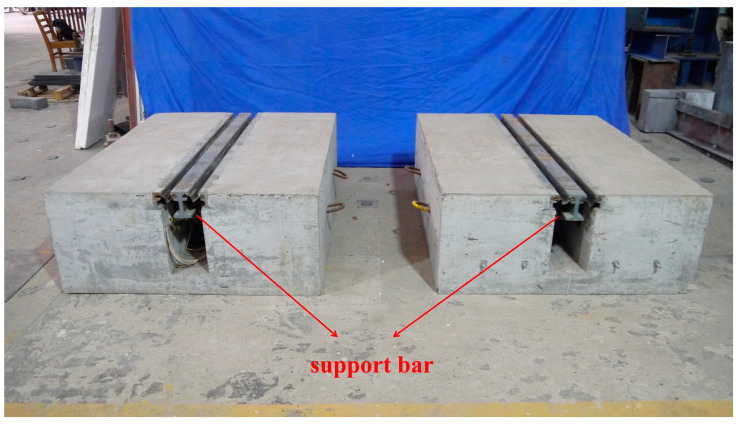
Two specimens for the experiment.

**Figure 5 sensors-23-03564-f005:**
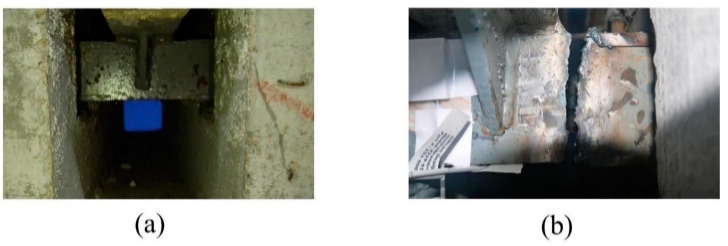
Support bar of the specimens: (**a**) healthy support bar; (**b**) damaged support bar.

**Figure 6 sensors-23-03564-f006:**
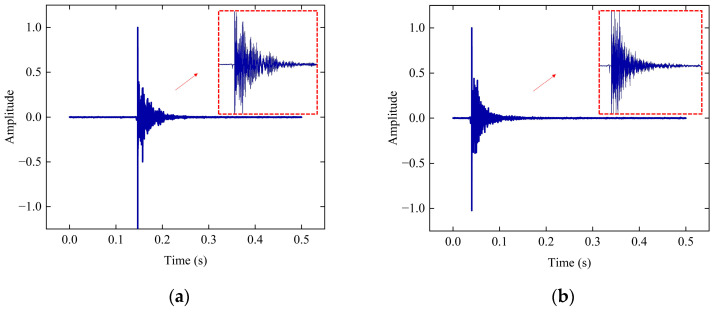
Time domain: (**a**) time domain of the damaged support bar; (**b**) time domain of the healthy support bar.

**Figure 7 sensors-23-03564-f007:**
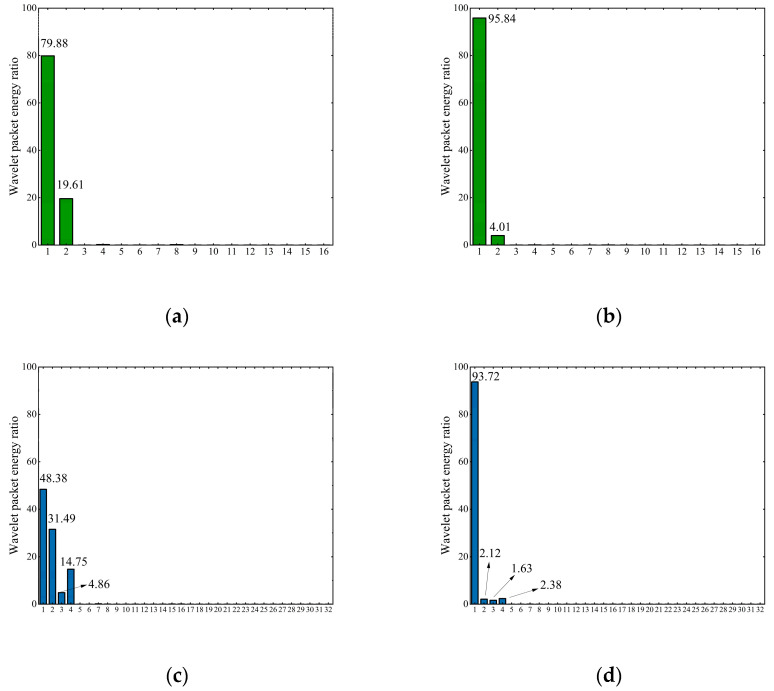
Wavelet packet energy ratio: (**a**) undamaged four-layer wave packet energy; (**b**) damaged four-layer wave packet energy; (**c**) undamaged five-layer wave packet energy; (**d**) damaged five-layer wave packet energy.

**Figure 8 sensors-23-03564-f008:**
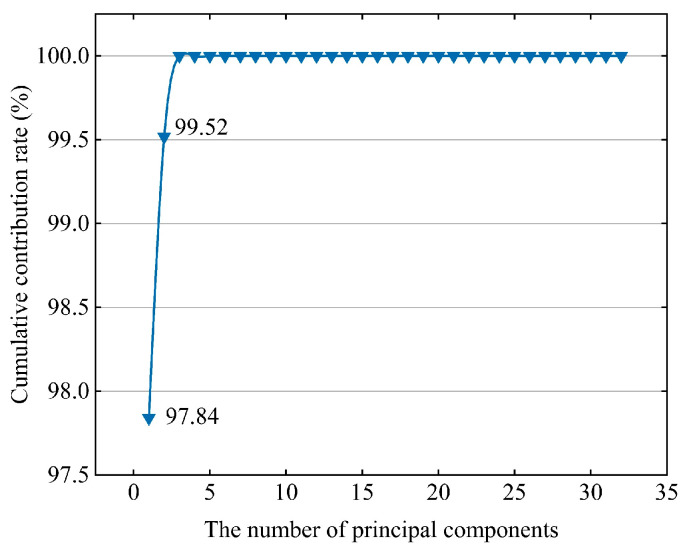
Results of the cumulative contribution rate.

**Figure 9 sensors-23-03564-f009:**
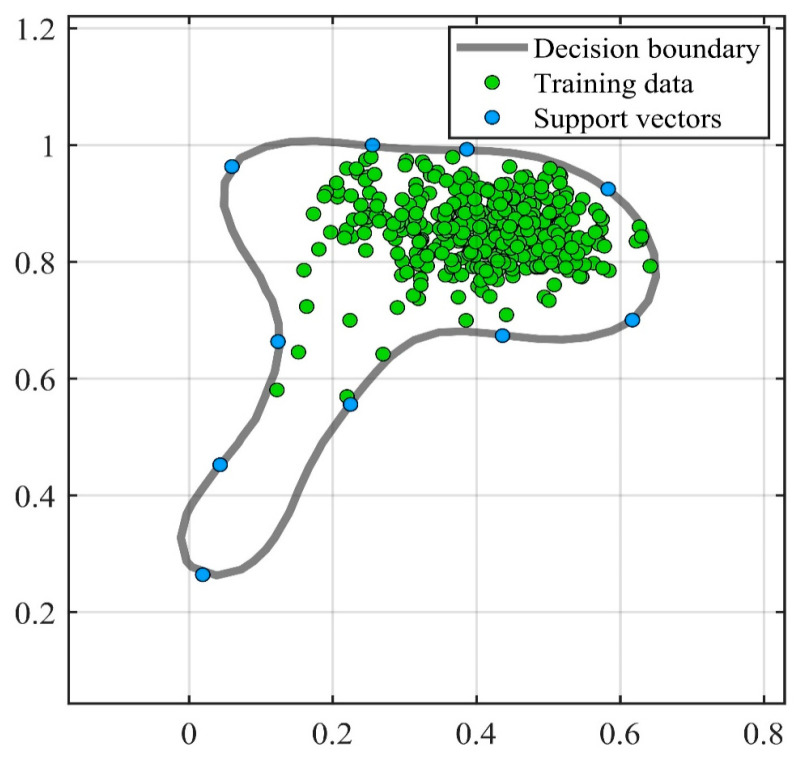
Results of training.

**Figure 10 sensors-23-03564-f010:**
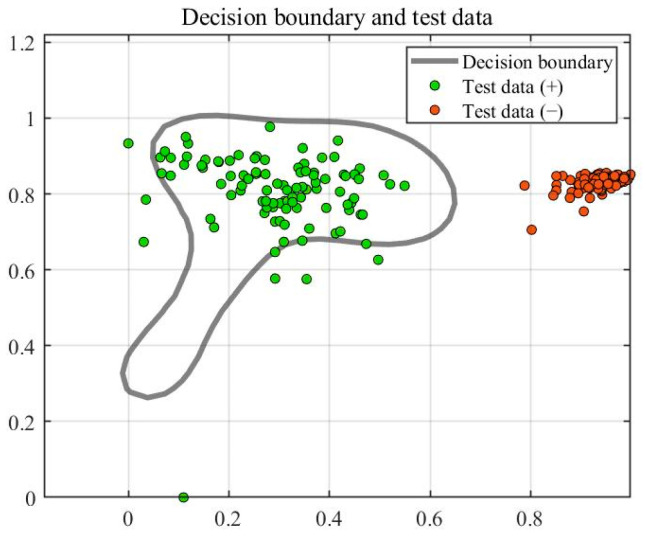
Results of testing.

**Figure 11 sensors-23-03564-f011:**
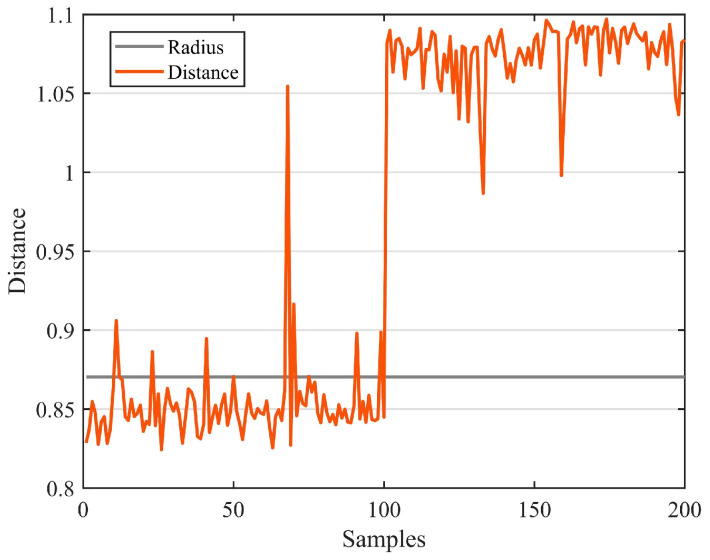
Identifying the results of damage.

**Figure 12 sensors-23-03564-f012:**
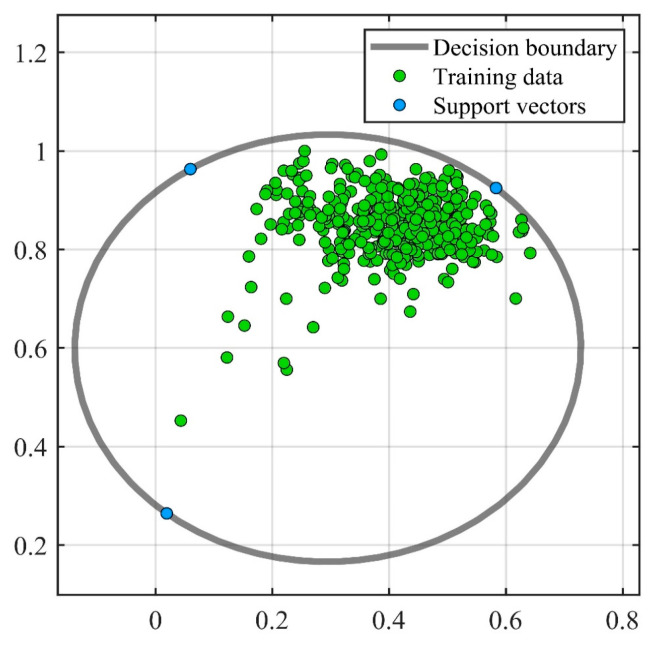
Training of BO−SVDD.

**Figure 13 sensors-23-03564-f013:**
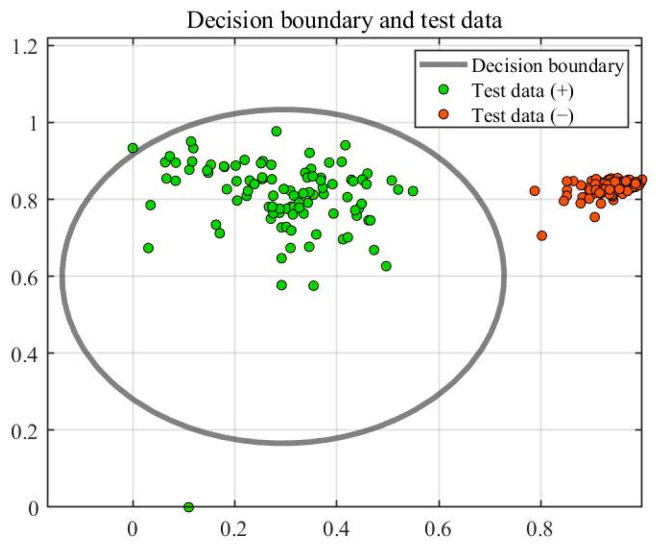
Testing of BO−SVDD.

**Figure 14 sensors-23-03564-f014:**
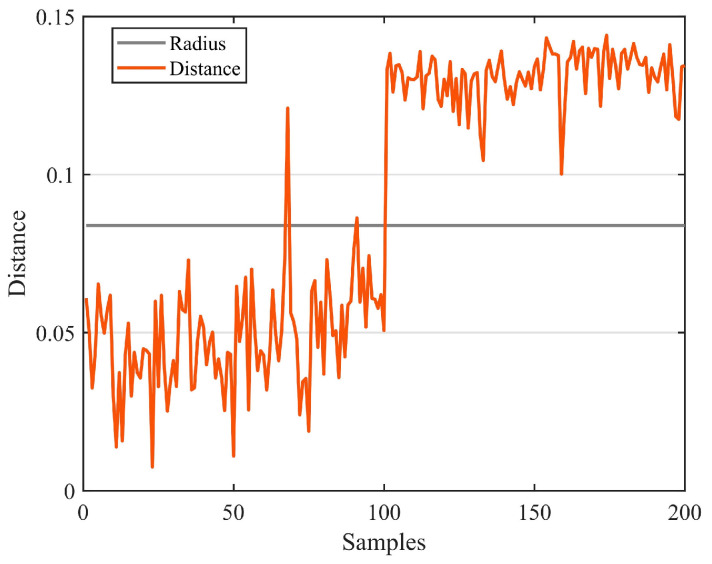
Identifying the results of damage of BO−SVDD.

**Table 1 sensors-23-03564-t001:** Material properties of Q345B.

Elastic Modulus (GPa)	Shear Modular (GPa)	Yield Strength (MPa)	Ultimate Strength (MPa)	Density (kg·m^3^)	Poisson Ratio
**200**	76.9	350	150	7850	0.3

**Table 2 sensors-23-03564-t002:** Design of concrete (kg/m^3^).

Cement	Flyash	Mineral Powder	Sand	Crushed Rock	Water Reducer	Water
390	76.9	80	831	920	9.2	150

**Table 3 sensors-23-03564-t003:** Parameters and results of the experiment.

	**Cost**	**Gamma**	**R**	**Time (s)**	**Accuracy**
**SVDD**	20	15	0.8704	0.1555	95%
**PSO-SVDD**	1	0.0156	0.0766	30.5979	99%
**GA-SVDD**	0.4571	0.2243	0.2858	30.2120	99%
**BO-SVDD**	0.7574	0.0188	0.0839	15.6018	99%

## Data Availability

Not applicable.

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
