# Peer review of "Sound Damage Detection of Bridge Expansion Joints Using a Support Vector Data Description"

_sensors, 2023, doi:10.3390/s23073564_

Round 1
Reviewer 1 Report
1. Introduction was not clear. The second paragraph should be reorganized. The significance of this manuscript should be rewritten.
2. More discussions should be added on comparing boundary conditions between tested specimens and real bridges.
3. Why were only material properties of steel included in Table 1? Do the material properties of concrete affect the tested result?
Author Response
Thank you for your letter and the reviewer’s comments concerning our manuscript entitled “Sound damage detection of bridge expansion joint using the support vector data description.” (sensors-2262107). Those comments are valuable and very helpful. We have read through the comments carefully and have made corrections. In addition, we have improved the paper using the official mdpi language modification service. Please see the attachment. Revised portions are marked in blue in the paper.

Reviewer 2 Report
Review of the paper sensors-2262107, entitled “Sound damage detection of bridge expansion joint using the support vector data description”, authored by Junshi Li, Caiqian Yang and Jun Chen.
(1) The paper reports the results of a preliminary study aimed at defining and testing an innovative technique for damage detection of bridge expansion joints through sound analysis. The influence of the damaged condition on the wavelet packet energy ratio is firstly assessed, and the detection technique is refined by implementing PCA and support vector data description assessment. The study proves that the techniques can be effective in detect the damage to support bar of the investigated bridge expansion joint.
(2) The paper investigates an issue of paramount interest, and the contents fall within the scope of the Journal. The manuscript is relatively well written, even though it can be improved. The investigated scenario is not easily generalizable or extendable but it represents in useful pilot application. The study potentially contributes to the field since it provides a methodology and technical and quantitative criteria for damage detection in bridge expansion joint, based on the analysis of the sound signals. However, revisions should be implemented in order to reach the adequate level of quality. The Authors are recommended to implement the revisions reported in the detailed reviewed manuscript report.

Author Response

(The authors gave the same response as above.)

Reviewer 3 Report
see the attachment

Author Response

(The authors gave the same response as above.)

Reviewer 4 Report
1. Line 20-21: Change to uniform cases.
2. Line 97-103: The method and process proposed in your paper are similar to the paper you quoted. What are the differences between you two? And what are the advantages of your method compared to others?
3. Line 108: Summarize the research status, not just list them. Please briefly outline the current shortcomings, and then present the research methodology of this paper.
4. Line 117: Introduce a more detailed layout of what you are delivering per section in this paper.
5. Figure 1: What does the preprocessing mentioned in the figure include? It is not reflected in the whole text.
6. Line 162: What is zj, is j the same as the j in Equation (4)?
7. Line 198-202: How to choose the parameters c1, c2, w?
8. Line 246, 250, 253: All figures and tables should be cited in the main text as Figure 1, Table 1, etc.
9. Figure 2: Please mark the position of the support bars in the figure.
10. Line 270-273: After reading the full text, I found that you did not analyze the noise immunity of the method. Noise is inevitable in the actual measurement environment. Please confirm the effectiveness of the proposed method in practical engineering.
11. Figure 5: Which position is this in Figure 4? Is it cross section or longitudinal section? Mark it in Figure 4.
12. Line 284-285: How do you define a healthy state? Is it the data obtained by hitting the healthy support bars once?
13. Line 290-292: “Therefore, the db6 basis function is applied to wavelet packet decomposition to the processed signal. Therefore, the db6 basis function is used to apply wavelet packet decomposition to the processed signal.” Please check if this part is duplicated.
14. Figure 6: “(a) Time domain of the healthy support bar; (b) Time signal of the damaged support bar.” uniform the description.
15. Line 302-303: “Therefore, this paper selects 5-layer wavelet packet decomposition to extract the feature vectors.” Is this conclusion only inferred from a set of healthy and damaged data?
16. Line 316-318: “The penalty parameter C is initially set to 20. The Gaussian kernel function is selected the kernel function, and the kernel parameters are initially set to 15.” Why do these two parameters take these two values?
Author Response

(The authors gave the same response as above.)

Round 2
Reviewer 2 Report
The Authors have improvede the paper according to the Reviewer's suggestions and have provided the clarifications to the Reviewer's questions. The paper, in the Reviewer's opinion, can be accepted for publication.
Reviewer 4 Report
As my earlier comments have been corrected and clarified, in my opinion, the article can be published in Sensors.